# Effector Profiles of Endophytic *Fusarium* Associated with Asymptomatic Banana (*Musa* sp.) Hosts

**DOI:** 10.3390/ijms22052508

**Published:** 2021-03-02

**Authors:** Elizabeth Czislowski, Isabel Zeil-Rolfe, Elizabeth A. B. Aitken

**Affiliations:** School of Agriculture and Food Sciences, The University of Queensland, St. Lucia, QLD 4072, Australia; elizabeth.czislowski@uqconnect.edu.au (E.C.); isabel.zeilrolfe@uq.net.au (I.Z.-R.)

**Keywords:** effectors, *SIX* genes, endophytes, *Fusarium*

## Abstract

During the infection of a host, plant pathogenic fungi secrete small proteins called effectors, which then modulate the defence response of the host. In the *Fusarium oxysporum* species complex (FOSC), the secreted in xylem (*SIX*) gene effectors are important for host-specific pathogenicity, and are also useful markers for identifying the various host-specific lineages. While the presence and diversity of the *SIX* genes has been explored in many of the pathogenic lineages of *F. oxysporum*, there is a limited understanding of these genes in non-pathogenic, endophytic isolates of *F. oxysporum*. In this study, universal primers for each of the known *SIX* genes are designed and used to screen a panel of endophytically-associated *Fusarium* species isolated from healthy, asymptomatic banana tissue. *SIX* gene orthologues are identified in the majority of the *Fusarium* isolates screened in this study. Furthermore, the *SIX* gene profiles of these endophytic isolates do not overlap with the *SIX* genes present in the pathogenic lineages of *F. oxysporum* that are assessed in this study. *SIX* gene orthologues have not been commonly identified in *Fusarium* species outside of the FOSC nor in non-pathogenic isolates of *F. oxysporum*. The results of this study indicate that the *SIX* gene effectors may be more broadly distributed throughout the *Fusarium* genus than previously thought. This has important implications for understanding the evolution of pathogenicity in the FOSC.

## 1. Introduction

Species within the *Fusarium* genus represent some of the most devastating and important pathogens of many of the world’s agricultural crops. The *Fusarium oxysporum* species complex (FOSC) includes isolates that can be pathogens, saprophytes and even endophytes [1,2]. The plant pathogenic lineages of *F. oxysporum* have been recorded to cause vascular wilt in over 100 species of plants [3,4,5]. However, an individual pathogenic isolate of *F. oxysporum* is a specialist and typically causes disease in only one or two host species. As a consequence of this host-specificity, isolates of *F. oxysporum* are classified into special groups called ‘*formae speciales*’ (plural, ff. spp.; singular–*forma speciaies*, f. sp.) according to their specific host [1]. Many *formae speciales* consist of multiple, genetically distinct clonal lineages or vegetatively compatible groups (VCGs) [3,6,7].

Fusarium wilt of banana is one of the most notorious examples of disease caused by *F. oxysporum* and is also known as Panama disease of banana [8,9]. The isolates that cause Panama disease in banana have historically been referred to as *F. oxysporum* f. sp. *cubense* (*Foc*). However, recent taxonomic revisions within the FOSC have divided isolates of *Foc* into eight different species [10,11]. The relationship between these newly described species and other methods of describing and classifying isolates of *Foc* have been summarised in Table 1. For clarity of communication and consistency, the synonym *Foc* shall be used throughout this study to indicate all isolates and species within the FOSC that are pathogenic to bananas. This convention for the naming of isolates *F. oxysporum* shall also be extended to the other *formae speciales* that have also been recently epitypified [11]. Isolates within *Foc* are further classified into cultivar-specific races and VCGs (Table 1) [3,6,7,12,13]. Presently, tropical race 4 (TR4; VCG 01213/16) is of significant concern as it is highly pathogenic to all commercial cultivars and is rapidly spreading throughout the world’s banana producing regions [14,15,16,17,18]. Phylogenetic analyses of the VCGs that comprise *Foc* have demonstrated that isolates within this *forma specialis* do not share a recent common ancestor and are commonly more closely related to isolates of *F. oxysporum* that are non-pathogenic or pathogenic to other host species [4,5,19,20,21]. It is hypothesised that the polyphyletic distribution of *Foc* in the FOSC is due to either convergent evolution and/or the horizontal transmission of the genes conferring host-specific pathogenicity. One such class of pathogenicity genes includes effectors.

Effector genes encode for small proteins that are secreted by plant pathogens, including *F. oxysporum*, during the infection of a host [23]. These proteins are often critical for infection as they manipulate the defence response of a host. In *F. oxysporum*, a family of effector genes termed secreted in xylem (*SIX*) genes have been identified [24]. Currently, 14 *SIX* genes have been experimentally identified in the tomato pathotype, *F. oxysporum* f. sp. *lycopersici* (*Fol*) [24,25,26,27,28]. Subsequent to their discovery, *SIX3*, *SIX4*, *SIX5* and *SIX6* have been experimentally verified to be required for full virulence of *F. oxysporum* in its respective host [24,29,30,31,32,33]. Furthermore, studies in the tomato and cucurbits have demonstrated that the small, dispensable accessory chromosomes to which the *SIX* genes are localised can be horizontally transferred from pathogenic lineages of *F. oxysporum* to a genetically distinct, non-pathogenic isolate of *F. oxysporum* [33,34]. In both the tomato and cucurbit pathosystems, the transfer of accessory chromosomes from a pathogenic donor into a non-pathogenic recipient was accompanied by a gain of pathogenicity towards the respective host. These experiments have been important for demonstrating the significant impact these accessory chromosomes and associated pathogenicity genes have on the phenotype of their host strain.

Although 14 *SIX* genes have been described in the *Fol* lineages, not all 14 *SIX* genes are carried by other *formae speciales* [22,30,35,36,37,38]. In fact, van Dam et al. demonstrated that the *formae speciales* of *F. oxysporum* each harbour a unique combination of effectors. These effector profiles can be used to distinguish the different host-specific lineages [39]. Furthermore, variation in the *SIX* genes has been particularly useful for distinguishing different races and clonal lineages that exist within a *forma specialis* [22,26,37,39,40]. The unique combination of effectors has been proposed to confer the host-specific pathogenicity exhibited by plant pathogens, such as *F. oxysporum*. Several *SIX* genes, including *SIX2* and *SIX6*, have been previously reported in other species of *Fusarium*, including *F. verticillioides*, *F. proliferatum*, *F. foetens*, *F. hostae* and *F. agapanthi* [36,41,42]. Homologues of *SIX1* and *SIX6* have also been described outside the Fusarium genus in *Colletotrichum orbiculare* and *C. higginsianum*, as well as *SIX1* in *Leptosphaeria maculans* [43,44,45].

In *Foc*, homologues of *SIX1* and *SIX9* have been identified in all isolates of *Foc*, while *SIX8* was only identified in the VCGs of race 4 of *Foc* [22,35,46]. The other *SIX* genes identified in *Foc* include *SIX2*, *SIX6*, *SIX7*, *SIX10*, *SIX13* and a pseudogeneised homologue of *SIX4* [22,35,46]. Specific homologues of the *SIX* genes were associated with the different races of *Foc*. Furthermore, Czislowski et al. demonstrated evidence of horizontal transfer of the *SIX* genes between the lineages of *Foc* [22]. Subsequently, it was hypothesised that the polyphyletic relationship of the *Foc* lineages can be attributed to the horizontal transference of effectors and pathogenicity genes, including the *SIX* genes [22]. Although the distribution and diversity of the *SIX* genes in the pathogenic lineages of *Foc* has been studied, less is known about the *SIX* genes in non-pathogenic lineages of *F. oxysporum* associated with banana and other healthy hosts.

There has been a strong emphasis on understanding the distribution and diversity of effectors in the plant pathogenic forms of *F. oxysporum*. Species of *Fusarium* are commonly isolated from the tissue of healthy, asymptomatic host plants, including banana [47,48,49,50,51]. For the purposes of this study, endophytically-associated non-pathogenic isolates are considered those that have been recovered from healthy, asymptotic host material. These isolates are to be considered non-pathogenic towards the host that they were recovered from—however, this should not automatically preclude them from possibly being pathogens of other hosts. The presence of effectors in non-pathogenic lineages of *F. oxysporum* is not as well understood. This is in part due to the difficulty in classifying an isolate of *F. oxysporum* as truly non-pathogenic. Although isolates of *F. oxysporum* are common in soil environments and in healthy, asymptomatic hosts, the classification of these isolates as truly non-pathogenic is practically impossible, due to the number of hosts that must be screened in pathogenicity assays [2]. The use of molecular methods to classify pathogenic lineages of *F. oxysporum* is further complicated by the polyphyletic nature of many of the *formae speciales* [5]. Therefore, the development of a molecular method that is capable of distinguishing pathogenic and non-pathogenic lineages of *F. oxysporum* is highly desirable. The majority of studies investigating the utility of the *SIX* genes for identifying and distinguishing lineages of *F. oxysporum* have predominantly focused on plant pathogenic *formae speciales*. Few studies have sought to better understand the frequency of *SIX* genes in isolates of *F. oxysporum* recovered from the soil of different ecological environments.

Isolates of *F. oxysporum* recovered from the rhizosphere or asymptomatic hosts have been shown to have a reduced number of effectors, including *SIX* genes. Rocha et al. identified *SIX* gene homologues in only 12% of *F. oxysporum* isolate recovered from the soil of native environments [52]. Inami et al. also screened isolates of *F. oxysporum* recovered from the soil or tissue of wild tomato species for the presence of *SIX1*, *SIX3* and *SIX4*. However, they was unable to detect any of these genes in any of the isolates [49]. Jelinski et al. found that the *SIX* genes were more frequently identified in isolates of *F. oxysporum* recovered from the soil of a tomato field known to be affected by *Fol* [53]. The isolates were screened for *SIX1–SIX7* and half of the isolates had the same *SIX* gene profile as *Fol*, while 3% of the isolates carried a combination of *SIX* genes that was not typical of *Fol*. Interestingly, one isolate that harboured the same *SIX* genes as *Fol* was non-pathogenic towards susceptible tomato in a subsequent pot trial. There are limited studies that have investigated whether *SIX* genes are present in endophytic, non-pathogenic isolates of *F. oxysporum* recovered from the internal tissue of healthy, asymptomatic hosts. It is also unclear if other *Fusarium* species associated with the same ecological niche as *F. oxysporum* also harbour *SIX* genes. Furthermore, although *Fusarium* species are commonly recovered from the tissue of banana hosts, it is unclear how frequently *SIX* genes are associated with these isolates and if any of the *SIX* genes in the endophytic isolates are similar to those found in the pathogenic lineages of *Foc*.

This study sought to ascertain if isolates of *Fusarium* associated with healthy, asymptomatic banana plants also carried any *SIX* gene homologues, and if so, was there any similarities to the *SIX* genes known to occur in pathogenic lineages of *Foc*. Similar to what has been observed for non-pathogenic lineages of the FOSC other *Fusarium* species, it was hypothesised that the *SIX* genes would be infrequently associated with isolates of endophytic, non-pathogenic *Fusarium* from the tissue of asymptomatic banana hosts. It was hypothesised that of the non-pathogenic, endophytic isolates identified to carry *SIX* genes, their *SIX* gene profile would be distinct from the *SIX* gene profiles of the pathogenic lineages of *Foc*.

## 2. Results

### 2.1. Universal Primers Successfully Detect SIX Genes in Pathogenic Formae Speciales

To enable the identification of the *SIX* genes in isolates of *F. oxysporum* associated with healthy banana hosts, universal primers capable of amplifying each of the fourteen *SIX* genes were developed. A single set of universal primers capable of amplifying all known homologues of *SIX9* could not be designed. The *SIX9* homologues were separated into two groups (*SIX9* (group 1) and *SIX9* (group 2)), based on their % identify to the *Fol-SIX9* sequence (Appendix A). The ability of the universal primers to amplify their target *SIX* gene was determined by screening isolates of *F. oxysporum* (*n* = 46) representing nine different *formae speciales*, whose *SIX* genes profiles could also be investigated with whole-genome sequencing [22]. This included 25 isolates of *Foc* representing VCGs 0120­01223 and all known races of *Foc*. Where possible, the same isolate that was used in the whole-genome sequencing projects was screened with the *SIX* gene PCR primers. Where possible multiple isolates from the same *forma specialis* were also screened with the universal primers. It was not possible to screen the exact isolates of *Fol* and cotton-infection *Fusarium oxysporum* f.sp. *vasinfectum* (*Fov)*, whose genomes have been assembled and made publicly available. In these instances, isolates from the same *formae speciales* held within the Brisbane Plant Pathology Herbarium (BRIP) were screened using the universal primers.

The performance of the primers was assessed by comparing the expected results from whole-genome sequencing to the amplicons produced by PCR using the universal primers. In the instances where the exact isolate used to generate whole-genome sequence information was screened by PCR, all primers produced *SIX* gene profiles identical to what was expected (Table 2). The isolate of *Fol* that was screened using the universal *SIX* gene primers was found to have an identical repertoire of *SIX* genes as predicted by the genome sequence data *Fol*-4287. However, the *SIX* gene profiles from the isolates of the cotton-infecting *Fov* that were screened in this study had several discrepancies with the *SIX* genes identified in the sequenced genome of *Fov*-NRRL25433. Firstly, although *SIX14* was absent in the genome of *Fov*-NRRL25433, all of the *Fov* isolates screened in this study were shown to carry an orthologue of *SIX14*. Secondly, an orthologue of *SIX9* (group 2) was identified in the genome of *Fov*-NRRL25543, yet was not detected by PCR in the isolates of *Fov* screened in this study. The distribution of *SIX6*, *SIX8* and *SIX11* was also shown to be present in the isolates of *Fov* included in this study, but absent in the reference *Fov*-NRRL25433 genome (Table 2). It has been previously established that Australian isolates of *Fov* are genetically distinct from other international isolates of *Fov* and have been hypothesised to have evolved within Australia from the endemic population of *F. oxysporum* [54]. Additionally, Chakrabarti et al. had previously demonstrated a difference in effector profiles between Australian and non-Australian lineages of *Fov* [55]. As a result, the differences in *SIX* gene profiles between the Australian *Fov* isolates screened by PCR and the reference genome of *Fov*-NRRL25433 was not unexpected. Therefore, the performance of the universal primers was determined to be satisfactory for identifying the *SIX* genes in the various *formae speciales* assessed in this study and proceeded to use the primers to screen isolates of *Fusarium* recovered from asymptomatic banana plants.

### 2.2. A Genetically Diverse Range of Fusarium Is Associated with Healthy Banana Tissue

To investigate the presence of *SIX* genes in endophytic *Fusarium* associated with banana, a collection of endophytic *Fusarium* species was generated by sampling and isolating 23 healthy, asymptomatic banana plants cultivated at two sites, Redlands (Queensland, Australia) and Mullumbimby (New South Wales, Australia). At both sites, plants with internal and external symptoms typical of Fusarium wilt were also identified and sampled (*n* = 5, Appendix A). Cultures derived from symptomatic plants were included in the subsequent analyses in parallel with the putatively endophytic cultures of *Fusarium* isolated from healthy, asymptomatic banana plants.

A total of 105 isolates of *Fusarium* were recovered from either root, pseudostem or peduncle tissue. Of these isolates, 25 cultures were derived from plants with internal and external symptoms of Fusarium wilt, while the remaining 80 cultures were isolated from healthy, asymptomatic hosts. From the Mullumbimby site, isolates within the FOSC and members of the *Fusarium fujikuroi* species complex (FFSC) were the most commonly isolated species identified by sequence analysis of the translation elongation factor 1-alpha (*EF1*-*α*) locus (Table 3). Isolates within the FOSC were frequently isolated from root tissue, while isolates of the FFSC were more commonly isolated from the pseudostem and peduncle tissue. Other species that were isolated from the Mullumbimby site included members of the *F. solani* species complex (FSSC) and *F. incarnatum-equiseti* species complex (FIESC). Similarly, from the Redlands site, isolates of the FOSC were also the most frequently isolated species and was more commonly isolated from root tissue (Table 3, Figure 1). A single isolate of both *F. fujikuroi* and *F. solani* were also recovered from plants grown at Redlands. A phylogenetic assessment of the genetic diversity of the *Fusarium* isolates was evaluated using the *EF1-α* sequence data.

A phylogenetic analysis was built from a 601 bp alignment with *F. delphiniodes* used as an outgroup for the phylogeny. Trees produced from Bayesian and maximum likelihood analysis both produced trees with similar topologies (Appendix A). The four species complexes of *Fusarium* that were identified through morphological and sequence analysis were each recognised as unique clades in the phylogenetic tree (Figure 1). Isolates of the FOSC represented the largest and most diverse species complex in the phylogenetic tree. Within the FOSC, three sub-clades were identified. The majority of the endophytic isolates and pathogenic *formae speciales* of the FOSC clustered into clade A and clade B. All isolates of *F. oxysporum* recovered from the Mullumbimby site clustered into clade A of the FOSC. As has been previously reported for many pathogenic lineages of *F. oxysporum*, several of the *formae speciales* were found to be polyphyletic. The putatively endophytic isolates of *F. oxysporum* clustered with several of the pathogenic isolates of *F. oxysporum*, including *F. oxysporum* f.sp. *medicaginis* (*Fomg*), *F. oxysporum* f.sp. *niveum* (*Fon*), *F. oxysporum* f.sp. *conglutinans* (*Focg*) and various VCGs of *Foc*. Isolates of *F. oxysporum* isolated from both symptomatic and asymptomatic plants clustered with VCGs if *Foc*, such as VCG0120, VCG0124 and VCG01214 (Figure 1). None of the isolates of *F. oxysporum* recovered in this study clustered with the tropical race 4 VCG 01213/16. Within the FFSC, three species were identified; *Fusarium sacchari*, *Fusarium proliferatum* and *Fusarium fujikuroi* with *F. sacchari* being the most common species. Interestingly, the isolates of *F. sacchari* recovered in this study and one reference sequence of *F. sacchari* were all placed into a single clade indicating a low degree of genetic diversity in these isolates (Figure 1). Several well-supported lineages were identified in the endophytes from the FIESC and FSSC. There did not seem to be any strong correlation between the genetic lineages of *Fusarium* identified from the analysis of the *EF1-α* and the tissue type from which the endophytes were isolated.

### 2.3. A High Proportion of Endophytic Fusarium Carry SIX Gene Orthologues

The assessment of the endophytic *Fusarium* for *SIX* genes showed that a high proportion of the isolates harboured one of more of the *SIX* genes. The PCR screens with the universal primers found that ~75% of isolates from Mullumbimby and 82% of the isolates recovered from Redlands were found to have one or more of the *SIX* genes (Table 3, Figure 2). Twenty-four of the endophytic *Fusarium* isolates (representing ~23 %) did not carry any *SIX* genes as while the rest of the isolates had between one and seven *SIX* genes (Figure 1). The most commonly identified *SIX* genes in the endophytic *Fusarium* sp. Were *SIX4*, *SIX7*, *SIX1*, *SIX2* and *SIX9* (group 1) (Figure 2). Orthologues of *SIX3*, *SIX5*, *SIX9* (group 2) and *SIX11* were not identified in any of the endophytic isolates.

### 2.4. Hierarchical Clustering Distinguishes Pathogenic and Non-Pathogenic Lineages of Fusarium

Prior to assessing the clustering patterns of the *SIX* gene profiles of the endophytic *Fusarium*, it was first determined whether the pathogenic *formae speciales* of *F. oxysporum* used in this study could be distinguished by their *SIX* gene profiles. A hierarchical cluster analysis, based on *SIX* gene presence/absence, showed that each of the *formae speciales* of *F. oxysporum* was distinguished from each other (Figure 3). Generally, isolates from the same *forma specialis* clustered together in a single clade, except for *Fov* and *Foc*. The Australian isolates of *Fov* were shown to form a single clade, however, this clade was distinct from the reference isolate of *Fov*, which was isolated from diseased cotton in China (Figure 3). While the *SIX* gene profiles of *Foc* was generally able to cluster lineages based on their cultivar-specific pathogenicity, the relationships between the lineages did not reflect the races of the *Foc* isolates. The *SIX* gene profile of VCG 01210 was shown to cluster more closely to the VCGs of tropical race 4, due to the presence of *SIX2* in this VCG. The VCGs 01214, 01221, 01222 and 0122 also did not cluster discretely with any of the clades representing race 1, race 2, subtropical or tropical race 4 VCGs. All of these VCGs were shown to have a reduced number of *SIX* genes in their profile. The hierarchical cluster analysis of the *formae speciales* of *F. oxysporum* demonstrated the utility of *SIX* gene presence/absence for distinguishing the various pathogenic lineages. The *SIX* gene profiles of the endophytic *Fusarium* were then incorporated into the hierarchical cluster analysis.

A hierarchical cluster analysis was used to assess and group the phyletic patterns of the *SIX* genes identified in the pathogenic *formae speciales* of *F. oxysporum* and the isolates of *Fusarium* recovered from banana in this study. The cluster analysis showed that the putatively endophytic isolates of *Fusarium* recovered from asymptomatic, healthy hosts had a *SIX* gene profile that was distinct from the pathogenic *formae speciales* of *F. oxysporum* (Figure 4). Several *formae speciales* clustered closely with some of the endophytic *Fusarium* isolates. Seven isolates of *Fusarium* formed a sister branch to *Foz*, due to the presence of *SIX7*. Several putatively endophytic *Fusarium* isolates were also shown to cluster with *Focg* and several VCGs of *Foc*. Eleven isolates of *F. oxysporum* recovered from symptomatic plants at the Redlands site were shown to have *SIX* gene profiles that were identical to subtropical race 4 of Foc (e.g., VCG 0120) and three isolates recovered from symptomatic plants at the Mullumbimby site were shown to have the same *SIX* gene profile of race 1/race2 of Foc (Figure 4). These isolates also clustered in the same clade as the isolates of either subtropical race 4 or race 1/race 2, respectively, in the *EF1-α*-based phylogeny. Due to the presence of external and internal symptoms of Fusarium wilt and the clustering of the isolates with pathogenic VCGs in both the *EF1-α* and *SIX* gene analysis, it was concluded that both *Foc* was present at both sites. Interestingly, the different species of *Fusarium* did not cluster into a single clade, but were instead dispersed throughout the different clades. Additionally, there did not seem to be any strong correlation between any of the *SIX* gene profiles of the endophytes and the tissue type or plant from which the endophytes were isolated from. Additionally, there are many examples throughout the dendrogram, demonstrating that endophytes from the two different sites shared an identical *SIX* gene profile.

## 3. Discussion

Plant pathogenic lineages of *F. oxysporum*, such as the banana pathogen *Foc*, represent a small fraction of the genetic diversity of the FOSC; many isolates exist in the rhizosphere as saprophytes whilst others can establish an endophytic, asymptomatic relationship with a host plant [2]. The distribution, diversity and evolution of effectors in the pathogenic lineages within the FOSC has been well-studied. However, the presence of effectors, such as the *SIX* genes, in putatively endophytic *F. oxysporum* is poorly understood.

Initially, this study hypothesised that isolates of *F. oxysporum* recovered from healthy, asymptomatic banana plants would carry no or very few *SIX* genes. A major finding of this study was the high frequency of *SIX* genes in endophytic isolates of *Fusarium* associated with asymptomatic banana hosts. Similarly, Jelsinki et al. had also reported the occurrence of *SIX* gene homologues in a high proportion of putatively endophytic isolates of *F. oxysporum* recovered from tomato producing fields [53]. These results are contradictory to Rocha et al. and Inami et al., which both reported a low or no incidence of *SIX* genes in putatively endophytic or saprophytic isolates of *F. oxysporum* [49,52]. The disparity between the results of these studies could be due to the land use and the environment from which the endophytes have been recovered. Both this study and Jelsinki et al. targeted endophytes in hosts being grown for agricultural purposes. In contrast, Rocha et al. and Inami et al. recovered putatively endophytic or saprophytic isolates of *Fusarium* from natural ecosystems or wild, uncultivated hosts. In intensive agricultural systems, effectors (such as the *SIX* proteins) could be important to the endophytic, as well as the pathogenic isolates of *Fusarium* as they may facilitate the growth and survival of the fungus as an asymptomatic biotroph. For endophytic *Fusarium*, the ability to better colonise and survive within their host may give the isolates a significant advantage of other fungi that are present in the same niche. Therefore, it is hypothesised that the high frequency of *SIX* genes in the endophytes of *Fusarium* may be the result of positive selection for endophytic isolates whose genomes include features, such as effectors that enable them to be better competitors in the ecological niche of a host plant, particularly in intensive agricultural systems.

There is increasing evidence that many mycorrhizal and endophytic species of fungi utilise effectors to successfully establish their mutualistic relationships with their hosts (reviewed by References [56,57]). The functions of three effectors in three mycorrhizal species have been well-studied with interesting results. The SP7 effector in the mutualist, *Rhizophagus irregularis*, and the MiSSP7 effector in *Laccaria bicolour* have both been shown to modulate plant hormone signalling pathways, thereby downregulating the host defence responses [58,59,60,61,62]. The increasing number of sequenced genomes representing mycorrhizal and endophytic species have shown that hundreds of effector candidates are predicted to be encoded by these species [63,64,65]. It is possible that similar to mycorrhizal species of fungi, the endophytic isolates of *F. oxysporum* also utilise effectors to suppress a host’s defence responses to facilitate the establishment of an intimate endophytic relationship. However, the studies to date that have investigated the presence of *SIX* genes in environmental isolates of *F. oxysporum* have not established if the *SIX* genes or other effectors are expressed or functional in these non-pathogenic interactions.

An alternative explanation for the increased frequency of *SIX* genes identified in the endophytic isolates compared to other studies is due to the primers used in this study. The universal primers developed in this study to amplify all *SIX* gene homologues that were known at the time of this could account for the increased frequency of *SIX* gene detection in isolates of *Fusarium*. The other studies, such as Inami et al., Rocha et al. and Jelinski et al., utilised primers that were commonly designed to amplify *SIX* genes from specific *formae speciales* [49,52,53]. While the use of PCR worked well for our purposes, it is an approach that is biased towards identifying known orthologues of previously identified *SIX* genes. The most unbiased approach for effector discovery and analysis would be whole genome sequencing. Currently, the majority of genomes of *F. oxysporum* that have been sequenced represent pathogenic lineages, and only two genomes of putatively non-pathogenic isolates of *F. oxysporum* are publicly available. The two sequenced genomes of non-pathogenic *F. oxysporum*, Fo47 and MN14, both show a reduced accessory genome [38,39]. These isolates were also shown by van Dam and Rep to have reduced numbers of a transposable element class termed mimps, which are associated with the promoter region of the *SIX* genes and other predicted effector candidates of *F. oxysporum* [28,42]. Interestingly, the genomes of *Foc* also shared similar hallmarks similar to the non-pathogenic isolates, including a small accessory genome and reduced number of identified mimps [38,39,42].

The cluster analysis of this study was also important for demonstrating that the *SIX* gene profiles of the putatively endophytic isolates were not the same as the *SIX* gene profiles of pathogenic *formae speciales* of *F. oxysporum* that were assessed in this study. Following the identification of the *SIX* genes in putatively endophytic isolates of *Fusarium*, it was hypothesised that endophytes from healthy, asymptomatic bananas should not have the same *SIX* gene profile as the pathogenic lineages of *Foc*. A hierarchical cluster analysis supported this hypothesis, and based on their *SIX* gene profiles, the majority of the endophytic isolates clustered separately from the pathogenic lineages of *F. oxysporum*. Several isolates of *F. oxysporum* recovered from Mullumbimby shared the same *SIX* gene profile as the race 1 VCGs of *Foc*, and some of the isolates recovered from Redlands clustered with the SR4 VCGs of *Foc*. These isolates also had an identical *EF1-α* sequence to the respective pathogenic VCGs of *Foc*. In both of these instances, the plants from which these isolates were recovered displayed both external and internal symptoms of Fusarium wilt at the time of sampling. Based on the *SIX* gene profiles and sequencing of the *EF1-α* locus, the race of *Foc* that was causing disease at both sites was identified, thus, demonstrating the utility of the *SIX* gene presence/absence typing.

The diagnostic potential of the *SIX* genes in *Foc* has also been utilised by Carvalhais et al. in the design and development of a molecular assay capable of distinguishing between the various races and vegetative compatibility groups (VCGs) of *Foc* [66]. Typically, molecular diagnostics are developed to be able to rapidly identify the intended target organism and return a binary ‘yes/no’ answer. This approach to molecular diagnostics is favoured for pathogens, such as TR4, due to the ability to design an assay that is sensitive, specific and fast. However, these methods of developing diagnostics have limitations, such as requiring an extensive prior understanding of the target pathogen. As a result, this approach to molecular diagnostics is often not effective and identifying or characterising novel pathotypes or even novel *formae speciales*. The future of molecular-based diagnostics of the *F. oxysporum* species complex lies in the ability to predict an isolate’s pathogenic potential from its genomic information. The diagnostic utility of the *SIX* genes and effectors more broadly has the potential to provide a novel means of pathogen identification and characterisation in the *F. oxysporum* species complex. Developing an improved understanding of effector presence/absence and evolution in the genetically diverse range of non-pathogenic and endophytic isolates of *Fusarium* will be critical to the future utility of an effector-based multi-locus sequencing typing of *Fusarium*.

In conclusion, this study has demonstrated that many of the *Fusarium* species isolated from healthy, asymptomatic banana tissue carry one or more *SIX* genes. Furthermore, the *SIX* gene profiles of isolates that are pathogenic to bananas have a complement of *SIX* genes that is distinct from their non-pathogenic counterparts. The results of this study could indicate that endophytic isolates of *Fusarium*, including *F. oxysporum*, could indicate that putative endophytes acquire and/or retain effectors to better colonise hosts and become more competitive in the ecological niche of a host banana plant. Although there may be a selective advantage for isolates that carry effectors, the endophytes may not be unable to cause disease in banana as they lack the full complement of molecular machinery required for virulence. Currently, the means by which putative endophytes acquire *SIX* genes is unknown and requires further investigation. While horizontal chromosome transfer has been reported in pathogenic lineages of *Foc,* it is not known whether *SIX* genes are also acquired by endophytes via HCT. This study is an important demonstration that endophytic *Fusarium* could be potential reservoirs of effector genes and could contribute to the evolution of novel pathotypes of *F. oxysporum*. For this reason, the endophytic and putatively non-pathogenic lineages of *F. oxysporum* continue to be of significant interest and warrant further research.

## 4. Materials and Methods

### 4.1. Universal Primer Design for SIX1–SIX14

The design of universal primers to amplify *SIX1–SIX14* was performed by identifying highly conserved sites suitable for primer design parameters in multiple sequence alignments of each gene constructed from sequence data available on GenBank (www.ncbi.nlm.nih.gov/genbank (accessed on 28 February 2021)). For each *SIX* gene, corresponding sequence homologues in the National Centre for Biotechnology Information (NCBI) nucleotide database and in the genomes of *Fusarium oxysporum* available on the whole genome sequencing database using BLAST were retrieved (version 2.7.0, National Centre for Biotechnology Information, Bethesda, MD, USA; accessed 17 January 2017). The coding sequences of the *SIX* genes from *Fol* were used as the query sequences (Appendix A). Sequences that aligned to the query sequence with an E-value of <0.05 were aligned using the online sequence alignment tool, MAFFT (version 7.307, Research Institute for Microbial Diseases, Osaka, Japan; mafft.cbrc.jp/alignment/server) using the G-INS-1 method. The resulting sequence alignments were imported into Geneious (version 6.8.1, Biomatter Pty. Ltd., AKLD, New Zealand; [67]) and were manually inspected and adjusted if required. Primer pairs that amplified all sequence homologues of a *SIX* gene were designed to highly conserved regions of the genes using the primer3 plugin within Geneious (version 6.8.1, Biomatter Pty. Ltd., AKLFD, New Zealand) (Table 4). The following parameters were used by primer3 to design the primers; regions with at least 75% identity, annealing temperature range of 57–63 °C with an optimal temperature of 60 °C, maximum of 5 °C difference between primer pairs, and the resulting primer pairs had no or low duplex potential. For *SIX6*, *SIX9* and *SIX13*, where primer3 was unable to identify suitable sites, primer pairs were manually designed to highly conserved regions with a maximum of 5 °C difference between primer pairs, and the resulting primer pairs had no, or low, duplex potential as assessed by primer3. The annealing temperature and duplex potential for manually designed primers were determined using the primer3 feature within Geneious (version 6.8.1, Biomatter Pty. Ltd., AKLD, New Zealand).

For *SIX9*, a single pair of primers could not be designed to amplify all sequence homologues of *SIX9*. Instead, the sequence homologues were grouped into either Group 1 or Group 2 (Appendix A). Generally speaking, homologues that had <75% homology to the reference sequence from *Fol*-4287 (GenBank accession XM_018394292.1) were considered Group 1 and the primers SIX9f-G1 and SIX9r-G1 were designed to amplify the homologues in this group. Those homologues that had >75% homology to reference sequence of *SIX9* from *Fol*-4287 were considered Group 2, and the primers SIX9f-G2 and SIX9r-G2 were designed to amplify the homologues in this group.

### 4.2. Culturing and Nucleic Acid Extraction

Monoconidial cultures of *F. oxysporum* were initiated from carnation leaf stocks onto ¼ potato dextrose agar (PDA) (Difco Laboratories, Becton Dickinson Diagnostics, East Rutherford, NJ, USA) and incubated in the dark at 24 °C for seven days. A mycelial plug from each culture was used to inoculate a ½ potato dextrose broth culture (PDB; Difco Laboratories, Becton Dickinson Diagnostics, East Rutherford, NJ, USA). The broth cultures were shaken for seven days under 12 h light/12 h dark cycles. Mycelia were harvested by filtering through Miracloth (Merck Merck & Co., Galloping Hill Road Kenilworth, NJ, USA) and snap frozen in liquid nitrogen. The harvested mycelia were ground in liquid nitrogen, and nucleic acid was extracted using the CTAB method described by Leslie and Summerell [68]. The DNA concentration and purity were determined using a Nanodrop spectrophotometer (ThermoScientific, Waltham, MA, USA) and diluted to ~50 ng/uL with nuclease-free water.

### 4.3. Optimisation and Screening of F. oxysporum Formae Speciales with Universal Primers for the SIX Genes

The universal primers targeting the *SIX* genes were validated and optimised against a panel of isolates belonging to *formae speciales* of *F. oxysporum* whose *SIX* gene profiles could be established from data available from NCBI. To first establish the *SIX* gene profiles of the *formae speciales* listed in Table 5, the genomes of the *formae speciales* that were available on the NCBI genome database were queried using sequence homologues of the *SIX* genes identified in *Fol* using BLAST (version 2.7.0, National Centre for Biotechnology Information, Bethesda, MA, USA) (Appendix A). To optimise the thermocycling conditions, the optimal annealing temperature for each primer pair was identified using a gradient PCR. The optimal annealing temperature was defined as the temperature at which non-specific amplicons were eliminated, while retaining efficient amplification of the target loci. Amplification of the *EF1-α* region using the primers, EF1 and EF2, described by O’Donnell et al., confirmed that the DNA was of suitable quality for PCR assays [19]. All PCRs were conducted in 25 µL reactions consisting of 12.5 µL Promega GoTaq Green Master Mix (2*X*; Promega Corporation, Fitchburg, WI, USA), 1 µL forwards primer (10µM; Integrated DNA Technologies Inc.), 1 uL reverse primer (10 µM; Integrated DNA Technologies Inc.), 1 uL of template DNA (~50 ng/uL) and 9.5 µL of nuclease-free H_2_O. PCR amplification reactions were conducted using a Bioline thermocycler thermocycler (Meridian Bioscience Incorporated, London, England). The thermocycling conditions were as follows: (i) An initial denaturation at 95 °C for 2 min followed by (ii) 35 cycles of 95 °C for 30 s, then an annealing temperature as detailed in Table 4 for 30 s, and an extension time as detailed in Table 4 at 72 °C, and finally, (iii) a final extension time of 72 °C for 5 min. Amplicons were size separated on a 1.5% agarose gel stained with ethidium bromide and visualised on a UV transilluminator. To confirm the specificity of the PCR amplifications, PCR amplicons were purified using the Promega Wizard PCR Purification Kit (Promega Corporation, Fitchburg, WI, USA) and sequenced at Macrogen (Macrogen Inc., Seoul, Korea) using the primers in Table 4. The resulting chromatograms were manually edited in Geneious version 6.8.1, Biomatter Pty. Ltd., AKLD, New Zealand) to remove low quality or ambiguous sequences. To confirm the sequence identity of the PCR amplicons, the sequences were used to query the nucleotide BLAST database. (version 2.7.0, National Centre for Biotechnology Information, Bethesda, MA, USA). The sequences were used in a multiple sequence alignment that also included the original *SIX* gene sequences used to develop the primers. The results of the universal *SIX* gene primer PCRs were cross-referenced to previous published *SIX* gene profiles for each isolate to identify any conflicting results.

### 4.4. Isolation of Fusarium Species Associated with Asymptomatic Bananas

To investigate the presence of *SIX* genes in *Fusarium* spp. associated with asymptomatic banana, sampling of banana plants with no external symptoms of infection by *Foc* was conducted at two locations. The first location was at the Redlands Research Station (Redlands, QLD, Australia). This site maintains several stands of the wild banana species, *Musa acuminata* subsp. *malaccensis*. Five plants that appeared healthy and with no external symptoms of wilting were chosen for sampling. At this site, an additional three plants showed internal and external symptoms of Fusarium wilt. Tissue was still collected from symptomatic plants for further analysis to determine if *Foc* could be identified. For each plant, several samples of pseudostem and root tissue were harvested for isolations. The second location was a commercial banana plantation that grew the banana cultivar, ‘Lady Finger’. Eighteen plants that appeared healthy and with no external symptoms of wilting were chosen for sampling. An additional two plants exhibited external and internal symptoms of Fusarium wilt disease. Tissue was isolated from this plant to determine if *Foc* could be identified. From these plants, tissue from the roots, lower pseudostem (~1 m from ground level), upper pseudostem (~3 m from ground level) and the peduncle of the plants was collected. Details of cultures isolated in this study are summarised in Appendix A.

The primary isolation of *Fusarium* spp. from the banana tissue was made by trimming excess tissue until the sections were ~3 cm × ~3 cm. Tissue segments were surface sterilised by immersing in a 1:10 hypochlorite solution for 60 s before rinsing in sterile distilled water and dried on sterile blotting paper. The sterilised tissue segments were then further aseptically dissected into pieces of ~1 cm lengths, and plated onto half-strength PDA (Difco Laboratories, Becton Dickinson Diagnostics, East Rutherford, NJ, USA) amended with 100 mg/mL of streptomycin (Sigma Aldrich, MO, USA). The primary cultures were incubated at 24 °C in dark conditions and checked daily for ~7 days for Fusarium-like growth according to features described in Leslie and Summerell [68]. Cultures that had morphological features typical of Fusarium species as described by Leslie and Summerell were subcultured onto water agar for single sporing [68]. The pure cultures were incubated at 24 °C for 5 days before mycelia were harvested for DNA extractions and PCR screens.

### 4.5. Phylogenetic Analysis

Amplicons from the *EF1-α* PCR were purified using the Promega Wizard PCR Product and Gel Clean Up Kit (Promega Corporation, WI, USA) and sequenced at Macrogen (Macrogen Inc., Seoul, Korea). To identify the species of the *Fusarium* endophytes the resulting *EF1-α* sequences were manually edited to remove low quality sequences in Geneious (version 6.8.1, Biomatter Pty. Ltd., AKLD, New Zealand)) and used as query sequences to perform BLAST searches of the NCBI nucleotide database (version 2.7.0, National Centre for Biotechnology Information, Bethesda, MA, USA), *EF1-α* dataset in FusariumID (http://isolate.fusariumdb.org/ (accessed on 28 February 2021)) and the *Fusarium* Multilocus Sequence Typing database (http://www.westerdijkinstitute.nl/fusarium/ (accessed on 28 February 2021)). The *EF1-α* sequences of *Fusarium* species were aligned with MAFFT (version 7.307, Research Institute for Microbial Diseases, Osaka, Japan; https://mafft.cbrc.jp/alignment/server/ (accessed on 28 February 2021)) using the G-INS-1 parameters. The resulting sequence alignment was manually inspected ambiguously aligned regions were trimmed. The *EF1-α* alignment was analysed using Bayesian inference using MrBayes (version3.2.7, Department of Biodiversity Informatics, Stockholm, Sweden [69]) using the GTR-G-I model of substitution. Two independent analyses were conducted for four Markov chain Monte Carlo (MCMC) chains for 2 000 000 generations. The analysis was sampled every 1000 generations with a burn-in of 25%. A maximum likelihood analysis (ML) with RAxML (version8.1, version 8.1, Scientific Computing Group, Heidelberg Institute for Theoretical Studies, Heidelberg, Germany; [70]) was also undertaken. Trees were visualised, and tree images were generated using iTOL (version 3.6, European Molecular Biology Laboratory, Heidelberg, Germany; [71]).

### 4.6. Screening of Fusarium Cultures for SIX Genes

DNA was extracted from the cultures of *Fusarium* isolated from banana tissue using the process previously described. To ensure that (i) the DNA was of sufficient quality for PCR, and (ii) the cultures were *Fusarium* species, amplification of a region in the *EF1-α* was conducted as previously described. Isolates that had an *EF1-α* amplicon of the expected size were further assessed for *SIX* genes using the optimised universal primers. Amplification of *SIX1–SIX14* was performed using the optimised conditions described above and in Table 4. Amplicons were size separated and visualised as previously described. The *EF1-α* and *SIX* gene amplicons of the expected sized were purified using the Wizard PCR and Gel Clean Up kit (Promega Corporation, WI, USA) and sequenced by Macrogen (Macrogen Inc., Seoul, Korea) using the primers used for amplification of the target gene. The resulting *EF1-α* and *SIX* gene sequences were deposited to GenBank (accessions MW076542-MW076821).

### 4.7. Sequencing Analysis of the SIX Genes

The chromatograms of the *EF1-α* and *SIX* gene amplicons were visualised in Geneious (version 6.8.1, Biomatter Pty. Ltd., AKLD, New Zealand) and manually edited to remove low quality or ambiguous sequencing results. To confirm the sequence identity of the *SIX* gene amplicons from the panel of *formae speciales* and from the primary cultures, the sequences were used to query the nucleotide BLAST database (version 2.7.0, National Centre for Biotechnology Information, Bethesda, MA, USA). The sequences were compared reference *SIX* gene sequences from other *formae speciales* by multiple sequence alignments and gene trees inferred using Bayesian inference with MrBayes (version 3.2.7, Department of Biodiversity Informatics, Stockholm, Sweden; [69]). The GTR-G-I model of substitution was implemented with two independent analyses for four Markov chain Monte Carlo (MCMC) chains. The number of generations was increased by 50,000 until the standard deviation of the split frequencies fell below 0.01. The analysis was sampled every 1000 generations with a burn-in of 25%. Trees were visualised, and tree images generated using iTOL (version 3.6, European Molecular Biology Laboratory, Heidelberg, Germany; [71]) (Appendix A).

### 4.8. Cluster Analysis of SIX Gene Profile

Hierarchical clustering of the *SIX* gene profiles was performed by creating a binary matrix where gene presence was indicated by a “1” and gene absence was indicated by a “0”. The gene presence/absence data matrix was used to calculate a Jaccard distance matrix in R with the ‘vegan’ package (version. 2.5-2; [72]), which was then used as input for hierarchical clustering with the average linkage in the ‘cluster’ package in R (version. 2.0.7-1; [73]).

## Figures and Tables

**Figure 1 ijms-22-02508-f001:**
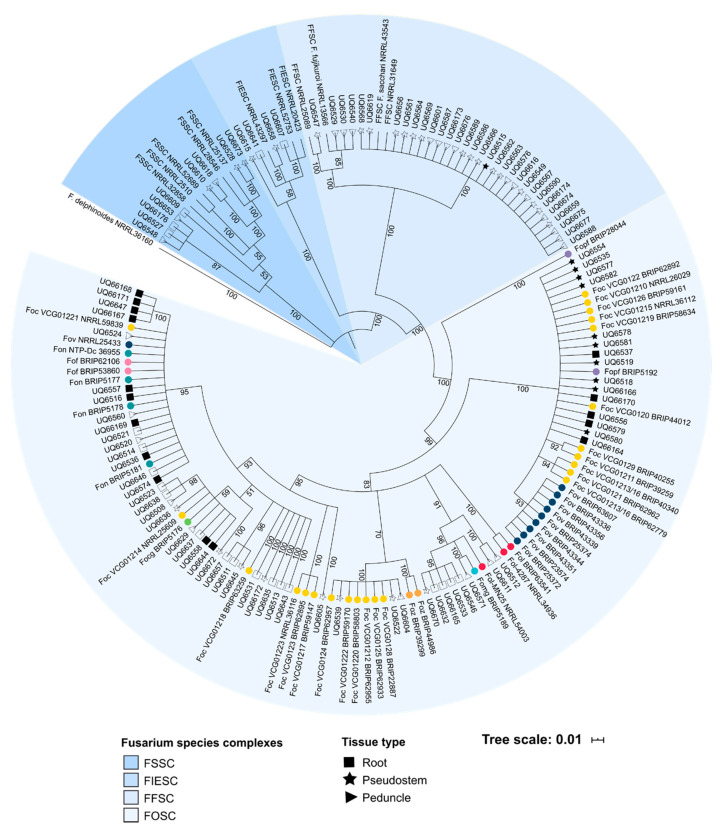
Phylogenetic tree of *Fusarium* isolates recovered in this study, reference sequences from pathogenic *formae speciales* of *Fusarium oxysporum* and other isolates from several *Fusarium* species complexes (FIESC—*Fusarium incarnatum-equiseti* species complex; FFSC—*Fusarium fujikuroi* species complex; FOSC—*Fusarium oxysporum* species complex; FSSC—*Fusarium solani* species complex). Pathogenic *formae speciales* of *F. oxysporum* are indicated with a coloured circle. The tissue from, which the isolates were recovered, is indicated by either a square (root), star (pseudostem) or triangle (peduncle). Isolates recovered from the Redlands site have a shaded shape. Isolates recovered from the Mullumbimby site have an unshaded shape. Nucleotide sequences from the translation elongation factor 1-α (*EF1-α)* locus were aligned and analysed using Bayesian inference. Branch labels indicate posterior probability determined during the analysis.

**Figure 2 ijms-22-02508-f002:**
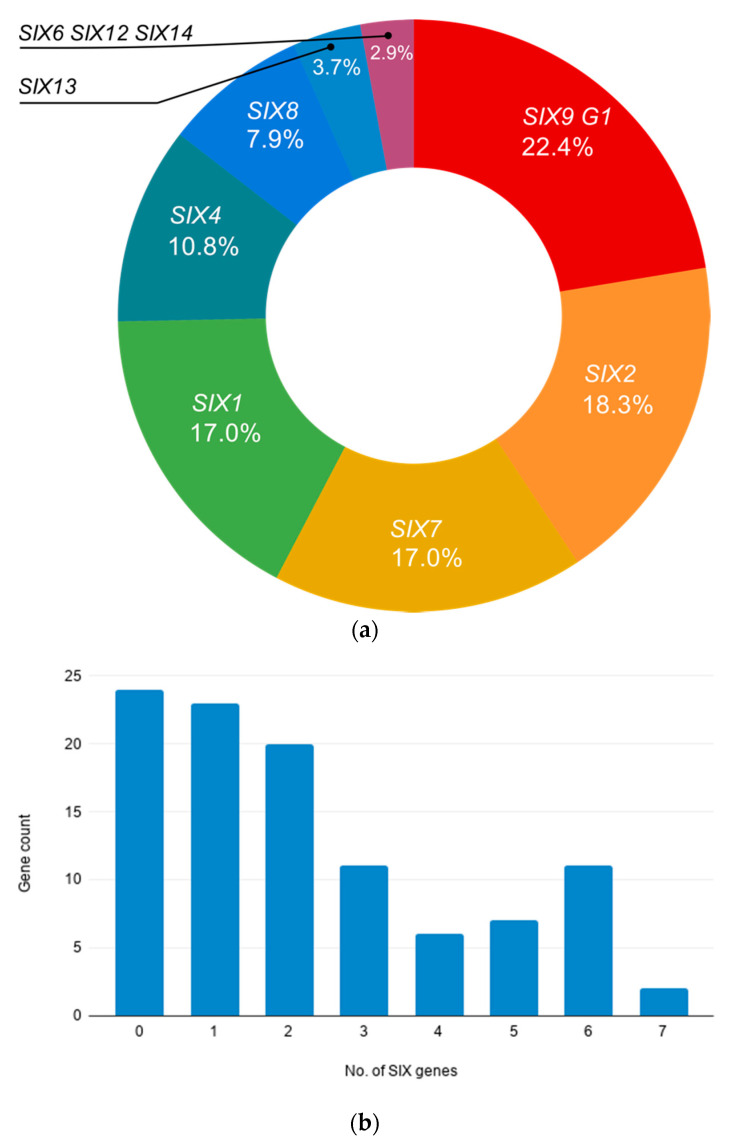
Summary of secreted in xylem (*SIX*) gene presence in isolates of *Fusarium* isolated in this study. (**a**) Frequency of *SIX* gene presence in isolates of *Fusarium* recovered from the tissue of field banana plants. (**b**) Summary of the number of *SIX* genes identified in isolates of *Fusarium* associated with healthy, asymptomatic banana plants.

**Figure 3 ijms-22-02508-f003:**
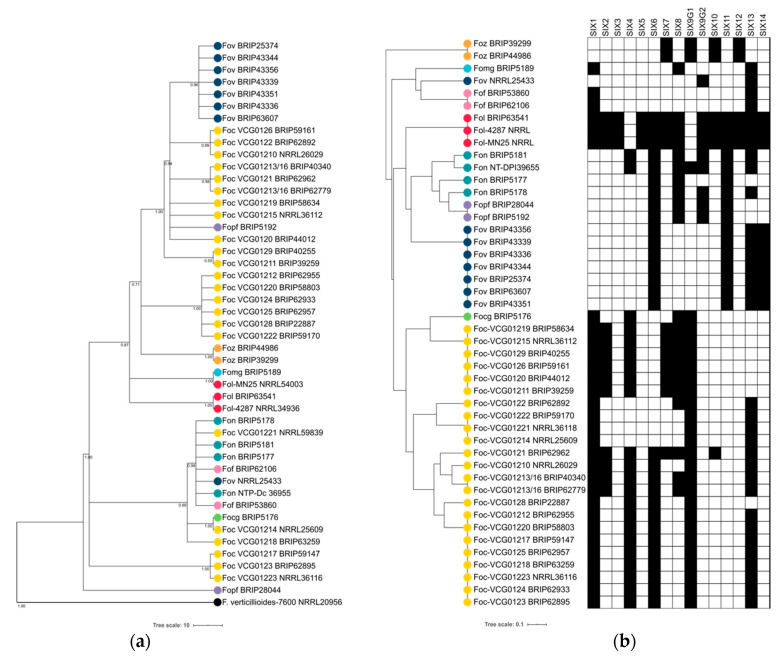
Hierarchical clustering analysis of the presence/absence of secreted in xylem (*SIX*) genes demonstrates a better association with the *formae speciales* of *Fusarium oxysporum* as compared to conventional phylogenetic analyses. (**a**) Genetic relationships of plant pathogenic *formae speciales* of *F. oxysporum* determined from an alignment and Bayesian analysis of the translation elongation factor 1-α (*EF1-α*). (**b**) Hierarchical clustering was used to analyse the *SIX* gene presence/absence matrix and generate a cluster dendogram for the *formae speciales* of *F. oxysporum*. The *SIX* gene presence/absence matrix used for the cluster analysis is given adjacent to the cluster dendrogram. A solid block indicates gene presence and a blank block indicates gene absence. Pathogenic *formae speciales* of *F. oxysporum* are indicated with a coloured circle.

**Figure 4 ijms-22-02508-f004:**
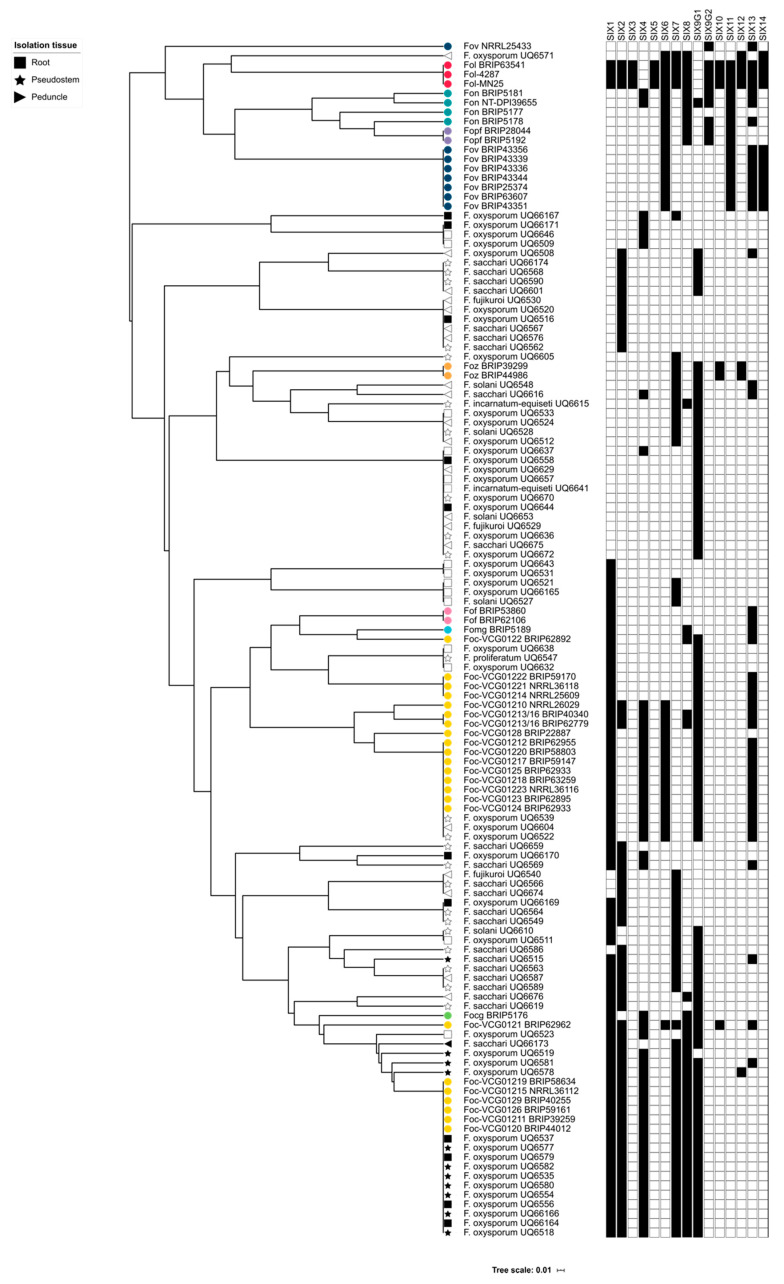
Hierarchical clustering of plant pathogenic *formae speciales* of *F. oxysporum* and *Fusarium* isolated from banana tissue in this study. The secreted in xylem (*SIX*) gene presence/absence matrix used for the cluster analysis is given adjacent to the cluster dendrogram in which a solid block indicates gene presence and a blank block indicates gene absence. Pathogenic *formae speciales* of F*. oxysporum* are indicated with a coloured circle. The tissue, from which the *Fusarium* isolates were recovered, is indicated by either a square (root), star (pseudostem) or triangle (peduncle). Isolates recovered from the Redlands site have a shaded shape. Isolates recovered from the Mullumbimby site have an unshaded shape.

**Table 1 ijms-22-02508-t001:** A summary of taxonomic species and other methods of developed to classify *Fusarium oxysporum* f.sp. *cubense*, including race and vegetative compatibility group (VCG) species designation within the *Fusarium oxysoprum* species complex (FOSC) [10,11,22].

Race	VCG	Species Name within FOSC
1	0123	*Fusarium phialophorum*
1	01210	*Fusarium purpurascens*
1, 2	0124	*Fusarium tardichlamydosporum*
1, 2	0125	*Fusarium tardichlamydosporum*
1, 2	0128	*Fusarium tardichlamydosporum*
1, SR4	01220	sp. (not determined)
2	01214	*Fusarium tardicrescens*
R4	0121	*Fusarium odoratissimum*
R4	0122	*Fusarium phialophorum*
SR4	0120	*Fusarium phialophorum*
SR4	0129	*Fusarium phialophorum*
SR4	01211	*Fusarium phialophorum*
SR4	0121	*Fusarium phialophorum*
SR4 ^1^	0126	*Fusarium purpurascens*
TR4	01213	*Fusarium odoratissimum*
TR4 ^2^	01216	*Fusarium odoratissimum*
Undetermined	01212	*Fusarium tardichlamydosporum*
Undetermined	01217	*Fusarium duoseptum*
Undetermined	01218	sp. (not determined)
Undetermined	01219	*Fusarium phialophorum*
Undetermined	01221	*Fusarium grosmichelli*
Undetermined	01222	*Fusarium tardichlamydosporum*
Undetermined	01223	*Fusarium duoseptum*
Undetermined	01224	*Fusarium duoseptum*
N/A ^3^	0127	N/A ^3^

^1^ Previously considered race 1. ^2^ Considered to be the same as VCG 01213. ^3^ VCG 0127 is no longer considered to be valid (VCG of race 3).

**Table 2 ijms-22-02508-t002:** A comparison of the secreted in xylem (*SIX*) gene profiles of the *formae speciales* in *F. oxysporum* by either querying reference genomes or screening by PCR with universal *SIX* gene primers developed for this study and confirmed by DNA sequencing.

*Forma Specialis* (Host Name) ^1^	Isolate ^2^	*SIX1*	*SIX2*	*SIX3*	*SIX4*	*SIX5*	*SIX6*	*SIX7*	*SIX8*	*SIX9*-G1 ^3^	*SIX9*-G2 ^3^	*SIX10*	*SIX11*	*SIX12*	*SIX13*	*SIX14*
*Fol* (tomato)	4287 NRRL34936^4^	+	+	+	-	+	+	+	+	-	+	+	+	+	+	+
*Fol* (tomato)	MN25 NRRL54003^4^	+	+	+	-	+	+	+	+	-	+	+	+	+	+	+
*Fol* (tomato)	BRIP63541	+	+	+	-	+	+	+	+	-	+	+	+	+	+	+
*Fopf* (passionfruit)	BRIP28044^5^	-	-	-	-	-	+	-	+	-	+	-	+	-	-	-
*Fopf* (passionfruit)	BRIP28044	-	-	-	-	-	+	-	+	-	+	-	+	-	-	-
*Fopf* (passionfruit)	BRIP5192	-	-	-	-	-	+	-	+	-	+	-	+	-	-	-
*Fon* (watermelon)	NT-DPI39655 ^4^	-	-	-	+	-	+	-	+	-	+	-	+	-	+	-
*Fon* (watermelon)	BRIP5178	-	-	-	-	-	+	-	+	-	+	-	+	-	+	-
*Fon* (watermelon)	BRIP5177	-	-	-	-	-	+	-	+	-	-	-	+	-	-	-
*Fon* (watermelon)	BRIP5181	-	-	-	+	-	+	-	+	-	+	-	+	-	+	-
*Fon* (watermelon)	NT-DPI39655	-	-	-	+	-	+	-	+	+	+	-	+	-	+	-
*Foz* (ginger)	BRIP39299 ^4^	-	-	-	-	-	-	+	-	+	-	+	-	+	-	-
*Foz* (ginger)	BRIP39299	-	-	-	-	-	-	+	-	+	-	+	-	+	-	-
*Foz* (ginger)	BRIP44986	-	-	-	-	-	-	+	-	+	-	+	-	+	-	-
*Fof* (strawberry)	BRIP53860^4^	+	-	-	-	-	-	-	-	-	-	-	-	-	+	-
*Fof* (strawberry)	BRIP53860	+	-	-	-	-	-	-	-	-	-	-	-	-	+	-
*Fof* (strawberry)	BRIP62106	+	-	-	-	-	-	-	-	-	-	-	-	-	+	-
*Fov* (cotton)	NRRL25433 ^4^	-	-	-	-	-	-	-	-	-	+	-	-	-	+	-
*Fov* (cotton)	BRIP63607	-	-	-	-	-	+	-	-	-	-	-	+	-	+	+
*Fov* (cotton)	BRIP43351	-	-	-	-	-	+	-	-	-	-	-	+	-	+	+
*Fov* (cotton)	BRIP25374	-	-	-	-	-	+	-	-	-	-	-	+	-	+	+
*Fov* (cotton)	BRIP43344	-	-	-	-	-	+	-	-	-	-	-	+	-	+	+
*Fov* (cotton)	BRIP43336	-	-	-	-	-	+	-	-	-	-	-	+	-	+	+
*Fov* (cotton)	BRIP43339	-	-	-	-	-	+	-	-	-	-	-	+	-	+	+
*Fov* (cotton)	BRIP43356	-	-	-	-	-	+	-	-	-	-	-	+	-	+	+
*Focg* (Brassicae family)	BRIP5176^4^	+	-	-	+	-	-	-	+	+	-	-	-	-	-	-
*Focg* (Brassicae family)	BRIP5176	+	-	-	+	-	-	-	+	+	-	-	-	-	-	-
*Fomg* (alfalfa)	BRIP5189 ^4^	+	-	-	-	-	-	-	+	-	-	-	-	-	+	-
*Fomg* (alfalfa)	BRIP5189	+	-	-	-	-	-	-	+	-	-	-	-	-	+	-
*Foc* (banana)	BRIP62933 ^4^ (VCG0124)	+	-	-	+	-	+	-	-	+	-	-	-	-	+	-
*Foc* (banana)	BRIP62933 (VCG0124)	+	-	-	+	-	+	-	-	+	-	-	-	-	+	-
*Foc* (banana)	BRIP62895 ^4^ (VCG0123)	+	-	-	+	-	+	-	-	+	-	-	-	-	+	-
*Foc* (banana)	BRIP62895 (VCG0123)	+	-	-	+	-	+	-	-	+	-	-	-	-	+	-
*Foc* (banana)	BRIP58698 ^4^ (VCG01217)	+	-	-	+	-	+	-	-	+	-	-	-	-	+	-
*Foc* (banana)	BRIP58698 (VCG01217)	+	-	-	+	-	+	-	+	+	-	-	-	-	+	-
*Foc* (banana)	NRRL36116 ^4^ (VCG01223)	+	-	-	+	-	+	-	-	+	-	-	-	-	+	-
*Foc* (banana)	NRRL36116 (VCG01223)	+	-	-	+	-	+	-	+	+	-	-	-	-	+	-
*Foc* (banana)	BRIP40340 ^4^ (VCG01213/16)	+	+	-	+	-	+	-	+	+	-	-	-	-	+	-
*Foc* (banana)	BRIP40340 (VCG01213/16)	+	+	-	+	-	+	-	+	+	-	-	-	-	+	-
*Foc* (banana)	BRIP62962 ^4^ (VCG0121)	+	+	-	+	-	+	+	+	+	-	+	-	-	+	-
*Foc* (banana)	BRIP62962 (VCG0121)	+	+	-	+	-	+	+	+	+	-	+	-	-	+	-
*Foc* (banana)	BRIP62892 ^4^ (VCG0122)	+	-	-	-	-	-	-	+	+	-	-	-	-	+	-
*Foc* (banana)	BRIP62892 (VCG0122)	+	-	-	-	-	-	-	+	+	-	-	-	-	+	-
*Foc* (banana)	NRRL36118 ^4^ (VCG01221)	+	-	-	-	-	-	-	-	+	-	-	-	-	+	-
*Foc* (banana)	NRRL36118 (VCG01221)	+	-	-	-	-	-	-	+	+	-	-	-	-	+	-
*Foc* (banana)	BRIP44012 ^4^ (VCG0120)	+	+	-	+	-	-	+	+	+	-	-	-	-	-	-
*Foc* (banana)	BRIP44012 (VCG0120)	+	+	-	+	-	-	+	+	+	-	-	-	-	-	-
*Foc* (banana)	BRIP63259 ^4^ (VCG01218)	+	-	-	+	-	+	-	-	+	-	-	-	-	+	-
*Foc* (banana)	BRIP63259 (VCG01218)	+	-	-	+	-	+	-	-	+	-	-	-	-	+	-
*Foc* (banana)	BRIP62779 ^4^ (VCG01213/16)	+	+	-	+	-	+	-	+	+	-	-	-	-	+	-
*Foc* (banana)	BRIP62779 (VCG01213/16)	+	+	-	+	-	+	-	+	+	-	-	-	-	+	-
*Foc* (banana)	BRIP39259 ^4^ (VCG01211)	+	+	-	+	-	-	+	+	+	-	-	-	-	-	-
*Foc* (banana)	BRIP39259 (VCG01211)	+	+	-	+	-	-	+	+	+	-	-	-	-	-	-
*Foc* (banana)	NRRL25609 ^4^ (VCG01214)	+	-	-	-	-	-	-	-	+	-	-	-	-	+	-
*Foc* (banana)	NRRL25609 (VCG01214)	+	-	-	-	-	-	-	-	+	-	-	-	-	+	-
*Foc* (banana)	NRRL26029 ^4^ (VCG01210)	+	+	-	+	-	+	-	-	+	-	-	-	-	+	-
*Foc* (banana)	NRRL26029 (VCG01210)	+	+	-	+	-	+	-	-	+	-	-	-	-	+	-
*Foc* (banana)	BRIP62933 ^4^ (VCG0125)	+	-	-	+	-	+	-	-	+	-	-	-	-	+	-
*Foc* (banana)	BRIP62933 (VCG0125)	+	-	-	+	-	+	-	-	+	-	-	-	-	+	-
*Foc* (banana)	BRIP59161 ^4^ (VCG0126)	+	+	-	+	-	-	+	+	+	-	-	-	-	-	-
*Foc* (banana)	BRIP59161 (VCG0126)	+	+	-	+	-	-	+	+	+	-	-	-	-	-	-
*Foc* (banana)	BRIP22887 ^4^ (VCG0128)	+	-	-	+	-	+	-	-	+	-	-	-	-	-	-
*Foc* (banana)	BRIP22887 (VCG0128)	+	-	-	+	-	+	-	-	+	-	-	-	-	-	-
*Foc* (banana)	BRIP40255 ^4^ (VCG0129)	+	+	-	+	-	-	+	+	+	-	-	-	-	-	-
*Foc* (banana)	BRIP40255 (VCG0129)	+	+	-	+	-	-	+	+	+	-	-	-	-	-	-
*Foc* (banana)	NRRL36112 ^4^ (VCG01215)	+	+	-	+	-	-	+	+	+	-	-	-	-	-	-
*Foc* (banana)	NRRL36112 (VCG01215)	+	+	-	+	-	-	+	+	+	-	-	-	-	-	-
*Foc* (banana)	BRIP59147 ^4^ (VCG01217)	+	-	-	+	-	+	-	-	+	-	-	-	-	+	-
*Foc* (banana)	BRIP59147 (VCG01217)	+	-	-	+	-	+	-	-	+	-	-	-	-	+	-
*Foc* (banana)	BRIP58634 ^4^ (VCG01219)	+	+	-	+	-	-	+	+	+	-	-	-	-	-	-
*Foc* (banana)	BRIP58634 (VCG01219)	+	+	-	+	-	-	+	+	+	-	-	-	-	-	-
*Foc* (banana)	BRIP58803 ^4^ (VCG01220)	+	-	-	+	-	+	-	-	+	-	-	-	-	+	-
*Foc* (banana)	BRIP58803 (VCG01220)	+	-	-	+	-	+	-	-	+	-	-	-	-	+	-
*Foc* (banana)	BRIP59170 ^4^ (VCG01222)	+	-	-	-	-	-	-	-	+	-	-	-	-	+	-
*Foc* (banana)	BRIP59170 (VCG01222)	+	-	-	-	-	-	-	-	+	-	-	-	-	+	-
*Foc* (banana)	BRIP62955 ^4^ (VCG01212)	+	-	-	+	-	+	-	-	+	-	-	-	-	+	-
*Foc* (banana)	BRIP62955 (VCG01212)	+	-	-	+	-	+	-	-	+	-	-	-	-	+	-

^1^ Fol—Fusarium oxysporum f.sp. lycopersici; Fopf—Fusarium oxysporum f.sp. passiflora; Fon—Fusarium oxysporum f.sp. niveum; Foz—Fusarium oxysporum f.sp. zingiberi; Fof—Fusarium oxysporum f.sp. fragariae; Fov—Fusarium oxysporum f.sp. vasinfectum; Focg—Fusarium oxysporum f.sp. conglutinans; Fomg—Fusarium oxysporum f.sp. medicaginis; Foc—Fusarium oxysporum f.sp. cubense. ^2^ NRRL—Agricultural Research Service Culture Collection, United States Department of Agriculture; BRIP—Brisbane Plant Pathology Herbarium; NT-DPI—Northern Territory Department of Primary Industries; VCG—vegetative compatibility group, as indicated next to isolates of Foc in parentheses. ^3^ SIX9-G1—SIX9 group 1; SIX9-G2—SIX9 group 2. ^4^ SIX gene profiles from these isolates were determined from whole-genome assemblies available on NCBI GenBank Genome database.

**Table 3 ijms-22-02508-t003:** Summary of *Fusarium* species isolated from the tissue of banana plants cultivated at two sites in New South Wales and Queensland, Australia.

Site	No. of Plants Sampled	Species Complex ^2^	Tissue Type			Total
			Root	Pseudostem	Peduncle	
Mullumbimby (NSW ^1^)	20	FFSC	0	16	14	30
		FIESC	2	2	0	4
		FOSC	17	8	9	34
		FSSC	3	3	3	9
		**TOTAL**	22	29	26	77
Redlands (QLD ^1^)	8	FFSC	0	1	N/A	1
		FIESC	0	0	N/A	0
		FOSC	17	10	N/A	27
		FSSC	0	0	N/A	0
		**TOTAL**	17	11	N/A	28

^1^ NSW—New South Wales, QLD—Queensland. ^2^ FFSC—*Fusarium fujikuroi* species complex, FIESC—*Fusarium incarnatum-equiseti* species complex, FOSC—*Fusarium oxysporum* species complex, FSSC—*Fusarium solani* species complex.

**Table 4 ijms-22-02508-t004:** PCR primers used in this study, including the expected amplicon size and annealing temperature for the primers. Primers for the translation elongation factor 1-α (*EF1-α*) were as described by O’Donnell et al. [19]. Universal primers for the *SIX* genes were all designed as part of this study.

Gene Target	Primers	Sequence (5′–3′)	Expected Amplicon Size (Base Pairs)	Annealing Temperature (°C)
*EF1-α*	EF1-F	ATGGGTAAGGARGACAAGAC	~650	55 °C
	EF2-R	GGARGTACCAGTSATCATGTT		
*SIX1*	SIX1f	TCT CCA TTA CTT TGT CTC ACG	694–733	58 °C
	SIX1r	CGA TTT AGG CGA TTC GGG G		
*SIX2*	SIX2f	GGT TCC CAT CGT TGA AGC	327–330	57 °C
	SIX2r	TTG GTT TAA ATC TGC GTG TC		
*SIX3*	SIX3f	TTA CTA CGA GCT TCA GCA CC	223	60 °C
	SIX3r	GCA TTA GGT GTT GCA ACA GG		
*SIX4*	SIX4f	CAG CTC AGA CAG TCA GCC	~491	58 °C
	SIX4r	GGC CTT GAG TCG AAT GAG C		
*SIX5*	SIX5f	TCA TCA GTA CTG TGC TTG CC	347–354	59 °C
	SIX5r	CAT GTT GAG TCT GCT CCT CC		
*SIX6*	SIX6f	CTC TCG AGA CAC SCT TCC	396–399	58 °C
	SIX6r	GAT CCA CCA ATA CCT TCA T		
*SIX7*	SIX7f	GAG GTG ACA TTT GAC ATC ACC	113	60 °C
	SIX7r	TAG TAT GCG CGC CAT TGG		
*SIX8*	SIX8f	CCC TAG CCG TCT CTG TGG C	163–165	64 °C
	SIX8r	CGT TCG ACA AGG GCT CTC TCG		
*SIX9* Group 1	SIX9f-G1	TTC AAG TCG GTT GCT ACG C	118	58 °C
	SIX9r-G1	GCA TCC CAA AAT CCA AAG CG		
*SIX9* Group 2	SIX9f-G2	CCG TCT TCT CTA CCG CCG	288	58 °C
	SIX9r-G2	AGT TGA CGC AAG CAA AGT CG		
*SIX10*	SIX10f	TCA CGT TTC GAG TTG GTC C	202	60 °C
	SIX10r	ACA CCA AAT CGA GTC GAT GC		
*SIX11*	SIX11f	GTT GCT CCT CCT TTG CTG G	163	62 °C
	SIX11r	TAC CAC TCT GAC CAG TCA CC		
*SIX12*	SIX12f	CAG AAT GCT TGT GTG TGT GG	171	61 °C
	SIX12r	ATC ACC AGA GCA TGA ACC CC		
*SIX13*	SIX13f	TCT GAT CAG CCT CCT AGC GT	840	60 °C
	SIX13r	CCA CTG TAA CTC GGC ATC GA		
*SIX14*	SIX14f	TGT CTC AGC GTA TCC TCG GC	147–197	61 °C
	SIX14r	ATT CAG TGA CAA CGG GAC CG		

**Table 5 ijms-22-02508-t005:** Isolates of *F. oxysporum* used to validate and optimise the universal secreted in xylem (*SIX*) gene primers. Abbreviated names of the *formae speciales* are indicated inside parentheses after the name of the *forma specialis*.

*Forma Specialis*	Accession Code ^1^	VCG ^2^	Host
*conglutinans* (*Focg*)	BRIP5176	NA	*Brassica oleracea* var. *capitate*
*cubense* (*Foc*)	BRIP62933	0124	*Musa* sp. (unidentified)
	BRIP62895	0123	*Musa* sp. AAB ‘Latundan’
	BRIP58698	01217	*Musa* sp. (unidentified)
	BRIP40340	01213	*Musa* sp. AAA ‘Cavendish’
	BRIP62962	0121	*Musa* sp. AA ‘Sucrier’
	BRIP62892	0122	*Musa* sp. AAA ‘Cavendish’
	NRRL36118	01221	*Musa* sp. ABB ‘Pisang Awak’
	BRIP44012	0120	*Musa* sp. AAA ‘Cavendish’
	BRIP63259	01218	*Musa* sp. (unidentified)
	BRIP62779	01216	*Musa* sp. AAA ‘Cavendish’
	BRIP39259	01211	*Musa* sp. AAB ‘Lady Finger’
	BRIP62911	0124	*Musa* sp. (unidentified)
	NRRL36113	01214	*Musa* sp. ABB ‘Bluggoe’
	NRRL25609	01214	*Musa* sp. ABB ‘Harare’
	NRRL26029	01210	*Musa* sp. AAB ‘Silk’
	BRIP62957	0125	*Musa* sp. (unidentified)
	BRIP59161	0126	*Musa* sp. (unidentified)
	BRIP22887	0128	*Musa* sp. ABB ‘Bluggoe’
	BRIP40255	0129	*Musa* sp. AAB ‘Lady Finger’
	NRRL36112	01215	*Musa* sp. AAA ‘Cavendish’
	BRIP59147	01217	*Musa* sp. (unidentified)
	BRIP58634	01219	*Musa* sp. (unidentified)
	BRIP58803	01220	*Musa* sp. (unidentified)
	BRIP59170	01222	*Musa* sp. (unidentified)
	BRIP62955	01212	*Musa* sp. (unidentified)
*fragariae* (*Fof*)	BRIP53860	NA	*Fragaria* × *ananassa*
	BRIP62107	NA	*Fragaria* × *ananassa*
	BRIP62106	NA	*Fragaria* × *ananassa*
*lycopersici* (*Fol*)	BRIP63541	NA	*Solanum lycopersicum*
	4287 ^3^ (NRRL34936)	0030	*Solanum lycopersicum*
	MN25 ^3^ (NRRL54003)	0033	*Solanum lycopersicum*
*medicaginis* (*Fomg*)	BRIP5189	NA	*Medicago sativa*
*niveum* (*Fon*)	NT-DPI36955	NA	*Citrullus* sp.
	BRIP5178	NA	*Citrullus lanatus*
	BRIP5177	NA	*Citrullus lanatus*
	BRIP5181	NA	*Citrullus lanatus*
*passiflorae* (*Fopf*)	BRIP28044	NA	*Passiflora edulis*
	BRIP5192	NA	*Passiflora edulis*
*vasinfectum* (*Fov*)	NRRL25433	0114	*Gossypium hirsutum*
	BRIP25372	01111	*Gossypium hirsutum*
	BRIP43365	NA	*Gossypium hirsutum*
	BRIP63607	NA	*Gossypium hirsutum*
	BRIP43351	NA	*Gossypium hirsutum*
	BRIP25374	01112	*Gossypium hirsutum*
	BRIP43344	NA	*Gossypium hirsutum*
	BRIP43336	NA	*Gossypium hirsutum*
	BRIP43339	NA	*Gossypium hirsutum*
	BRIP43356	NA	*Gossypium hirsutum*
*zingerberi* (*Foz*)	BRIP39299	NA	*Zingiber officinale*
	BRIP44986	NA	*Zingiber officinale*

^1^ BRIP—Brisbane Plant Pathology Herbarium, NT-DPI—Northern Territory Department of Primary Industries, NRRL—Agricultural Research Service Culture Collection, United States Department of Agriculture. ^2^ VCG—Vegetative Compatibility Group. NA was assigned to isolates where VCGs were unknown. ^3^ strain number commonly used for these isolates.

## Data Availability

The data presented in this study are openly available in GenBank at MW076542-MW076821.

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
