# Peer review of "Effector Profiles of Endophytic Fusarium Associated with Asymptomatic Banana (Musa sp.) Hosts"

_ijms, 2021, doi:10.3390/ijms22052508_

Round 1

Reviewer 1 Report

In the present paper by Czislowki et al. the authors characterize at a genetic level the presence and diversity of the SIX genes in different isolates of F. oxysporum obtained from banana plants in two different locations. They identify SIX gene orthologues in the vast majority of the Fusarium isolates screened throughout study.

Despite the through characterization of SIX gene presence in both pathogenic and “supposedly” non-pathogenic Fusarium isolates is an interesting topic, the rationale behind this study is not completely clear to me and some of the conclusions drawn by the authors are not supported by the data shown.

For example, Czislowki et al. claim that they analyse a panel of endophytically-associated Fusarium species isolated from healthy, asymptomatic banana tissue. However, based on the SIX gene repertoire found in some of these isolates they retract their nature later in the manuscript (line 319-322 and 417-418) and hypothesize that some of them are pathogenic since wilt symptoms were found on the plants from which they were isolated.

They also claim that SIX gene profiles of endophytic isolates do not overlap with the SIX genes present in the pathogenic lineages of F. oxysporum, however two major objections can be raised to this point:

  • The repertoire of SIX genes is highly variable inside each of the two groups
  • Not enough pathogenic isolates were screened in comparison to the “claimed” endophytic ones to suffice for a true comparison

Thus, in my opinion this study remains a mere characterization of SIX genes in different Fusarium isolates, which has already been performed, although not so extensively, in previous reports.

Given the nature of these weak points that exist throughout the manuscript I cannot recommend it for publication in ijms.   

Other points:

1)            Introduction and Conclusion paragraphs can be shortened, and redundant concepts could be merged (see for example the last two sections of the Introduction).

2)            As the authors mention in the text, SIX genes identified in the cotton infecting Fov isolate are quite different from those expected from the genome data of FOVNRRL25543. This is quite striking to me. Did they sequence or perform PCR reactions on the FOVNRRL25543 genomic DNA to ascertain that their degenerate primers are properly working?  

3) Table legends should be self-explanatory. Please specify how each formae specialis is abbreviated. In table 2 I would also add a column saying if each isolate is pathogenic or not, this would greatly ease the interpretation of results.

4) What is exactly shown in figure 2b? What is represented on the Y axis? If no isolates have between 8 and 14 SIX genes It is not useful to represent that data in the graph.  

5) In Supplementary Figure 1 legend, citation is not formatted.

Reviewer 2 Report

In the manuscript untitled ‘Effector profiles of endophytic Fusarium associated with asymptomatic banana (Musa sp.) hosts’  the authors used primers to target the SIX genes to screen a panel of endophytically-associated Fusarium species isolated from healthy, asymptomatic banana tissue. The manuscript is well written, and the relevance of the study is clear.

However, the manuscript revealed some points that I would like to be clarified but the authors.

Some authors of the present manuscript participate on a recent publication on the same subject (‘Molecular Diagnostics of Banana Fusarium Wilt Targeting Secreted-in-Xylem Genes’), in which it is presented diagnostic assays based on conventional PCR targeting SIX genes. The methodology presented on that paper allows to distinguish races and VCGs, I believe that some pathogenic and other endophytic. Although following a different approach, in the present paper the authors used primers to screen a panel of endophytically-associated Fusarium species isolated from healthy banana tissue. The authors should make reference to this previous publication, clarifying what is the additional knowledge of the methodology now presented.

My definition of degenerated primers is a mix of oligonucleotide sequences in which some positions contain several possible bases, giving a population of primers with similar sequences that cover all possible nucleotide combinations for a given protein sequence. Taking this definition in consideration, I do not understand why the authors refer to the design of degenerated primers to isolate SIX genes. In my opinion only EF primers are degenerated. I would like to have authors opinion on this subject.

Line 494-496: it seems that the sentence is not complete. Probably missing ‘were retrieved’

Reviewer 3 Report

The manuscript provides additional useful data on SIX genes in non- pathogenic Fusarium spp. isolates, obtained from banana plants. Based on available genome data and on a robust amount of PCR runs carried out, the study increases the actual knowledge on these effector proteins, and may be of interest for a broad audience of readers. The SIX genes data produced have potential either to understand the Fusarium spp. evolutive adaptation to parasitism or endophytism, and as possible detection target.

There are, however, a number of amendments needed before publication. In general the English style must be improved, as the text appears more the adaptation of a dissertation rather than a specific scientific report. Figures must be improved. In Figure 1, the resolution must be increased, the top left legend must be reduced increasing the dimension of the circle tree, also  moving the scale at the lower right angle. In Suppl. Fig. 1 in legend: use "identified with BLAST" in first line; indicate the BLAST genome reference number; if the reference sequence Fol-007 in the plot is also Fol-4287, it is suggested for clarity to add in legend also the latter, in parentheses before "indicated...", otherwise use FOl-007 in the figure instead; also add "Supplementary" before "Figure 1".

In Figure 2a: increase the size of characters in the pie chart, use blank ink on dark colors, enlarge the pie. Figure 2b: erase unnecessary empty parts (set axe y to 0-25 and axe x to 0-9). Figure 3: erase the contour lines surrounding the two dendrograms.

Lines 263-270 = move to Discussion. Also, try to reduce the Discussion, that is too long and verbose. See editing changes on the attached manuscript. Once these amendments are performed, the paper may be accepted for publication.

Round 2

Reviewer 1 Report

Despite the major amendaments made by the authors in both presenting the text and the figures/tables of their article I still feel the manuscript doesn't advances sufficiently (neither mechanicistically nor genetically) our current knowledge of the six genes repertoire in Fusarium oxysporum. 

As it stands, I cannot recommend this manuscript for publication in a high impact journal such as ijms. A completely different and focused setup, together with new data and insights would be necessary.